# Mechanical Properties and Microstructural Evolution of Ti-25Nb-6Zr Alloy Fabricated by Spark Plasma Sintering at Different Temperatures

Qing Zhu [1], Peng Chen [1], Qiushuo Xiao [1], Fengxian Li [1,*], Jianhong Yi [1,*], Konda Gokuldoss Prashanth [2,3,4] and Jürgen Eckert [2,5]

[1]  School of Materials Science and Engineering, Kunming University of Science and Technology, Kunming 650093, China
[2]  Erich Schmid Institute of Materials Science, Austrian Academy of Sciences, Jahnstraße 12, A-8700 Leoben, Austria
[3]  Department of Mechanical and Industrial Engineering, Tallinn University of Technology, Ehitajate tee 5, 19086 Tallinn, Estonia
[4]  CBCMT, School of Engineering, Vellore Institute of Technology, Vellore 632014, Tamil Nadu, India
[5]  Department of Materials Science, Montanuniversität Leoben, Jahnstraße 12, A-8700 Leoben, Austria
*  Correspondence: clfxl@kmust.edu.cn (F.L.); pnkgzyl@163.com (J.Y.); Tel.: +86-182-1390-3681 (F.L.)

**Abstract:** High-energy ball milling and spark plasma sintering (SPS) are used to create high-strength Ti-25Nb-6Zr biomedical alloys with β structures. The Ti-25Nb-6Zr alloy microstructure and mechanical properties were examined as a function of the sintering temperatures. The results showed that as the sintering temperature was raised, the densification process was expedited, and the comprehensive mechanical characteristics increased at first, then dropped slightly. Moreover, under high temperatures, the fracture morphology of the Ti-25Nb-6Zr biomedical alloys exhibited more dimples, indicating enhanced plasticity of the material. Evaluating the mechanical properties of the Ti-25Nb-6Zr biomedical alloy sintered at 1623 K indicated a high compressive strength of $1678.4 \pm 5$ MPa and an elongation of $12.4 \pm 0.5\%$. The strengthening mechanisms are discussed in terms of the formation and distribution of bcc-Ti in the matrix as well as the homogeneous distribution of Nb and Zr. This research presents a new method for fabricating Ti-25Nb-6Zr biomedical alloys with high strength and low modulus values. The theoretical grounds for the development of high-performance Ti-Nb-Zr alloys will be laid by detailed research of this technology and its strengthening mechanisms.

**Keywords:** Ti-Nb-Zr alloy; spark plasma sintering; powder metallurgy; microstructure; mechanical properties

## 1. Introduction

Humans' demands for biomedical implant materials have risen with the advancement of society and medical science. Because of their outstanding antifatigue performance and biocompatibility, medical implant materials such as titanium and titanium alloys are particularly well-suited for bone restoration in load-bearing sections [1]. In particular, titanium-based biomaterials with β-Ti-type structures with additions of Nb and Zr are suitable as medical implant materials [2,3] and have become a hot spot in the current research due to their good osteoblast adhesion [4,5]. There have also been three stages in the evolution of titanium alloys: the first stage focused on pure titanium and Ti-6Al-4V [6,7]. Ti-6Al-4V is mostly made up of α phase and β phase, and it has a moderate tensile strength of 893~931 MPa and a yield strength of 826~868 MPa [7]. When compared to pure titanium, Ti-6Al-4V performs better but has a higher elastic modulus than pure titanium, making its biocompatibility poor. Because it contains components such as Al and V that are hazardous to the human body, researchers are committed to studying other biological alloys that can

replace it, but at present, Ti-6Al-4V alloy is still widely used. [8,9]. The second stage was titanium alloys for vanadium-free implants, whose representative alloys are Ti-5Al-2.5Fe and Ti-6Al-7Nb [10,11]. Although the tensile strength and yield strength of Ti-6Al-7Nb are higher than those of the first-generation medical titanium alloys, its high modulus of elasticity is not close to that of human bone [10]. Although it is harmful to the human body to replace V with Fe and Nb in the Ti-6Al-4V alloy, the existing Al element will release Al ions after being implanted in the human body, which is also hazardous to human health [10]. The third stage of alloy development uses medical implant materials made of new vanadium-free and aluminum-free β-type titanium alloys that have a rather low elastic modulus values and are biocompatible. β-type titanium alloys have superior comprehensive mechanical characteristics and better biocompatibility than α-type titanium alloys [12]. One representative alloy is Ti-13Nb-13Zr [13]. Ti-13Nb-13Zr exhibits high tensile and yield strengths of 972~1038 MPa and 834~906 MPa, respectively, and, in addition, has a low Young's modulus of 80~85 GPa, making it more similar to the human skeleton than other metal and alloy implants now in use [13]. It was also found that Nb and Zr are nontoxic elements that are beneficial to cell proliferation and differentiation [14]. As a result, the development of new β-type titanium alloys is one of the key research hotspots in biometal materials research.

Currently, a number of investigations have described the synthesis of Ti-Nb alloys using vacuum arc melting technology, deformation, and subsequent heat treatment. Sun et al. [15] used cold rolling and heat treatment to obtain ultrafine grains and nanocrystalline α and ω phases with grain sizes of 1~2 μm, thereby improving the mechanical properties of the material. Zhou et al. [16] prepared $Ti_{40}Zr_{25}Ni_8Cu_9Be_{18}$ metallic glass through copper mold casting at different liquid phase temperatures and improved the plasticity of the material through crystallization. However, there are certain drawbacks to these approaches, such as the ease with which grain growth can occur and the ease with which composition segregation can occur when the melting temperatures of each component differ substantially, affecting the alloys' overall mechanical performance. Combining mechanical alloying with spark plasma sintering allows the possibility to effectively increase the mechanical characteristics and elastic modulus values of titanium-based bioalloys. For example, Xu et al. [17] created a Ti-43Al-9V alloy with tiny grains following this approach. Mechanical alloying and spark plasma sintering were also utilized by Zou et al. [18] to create an ultra-fine-grained Ti-35Nb-7Zr-5Ta alloy. Its microstructure contained body-centered cubic β-Ti surrounded by a hexagonal close-packed α-Ti structure. Wen et al. [19] prepared a Ti-26Nb-5Ag alloy by three methods (mechanical alloying, vacuum furnace sintering, and spark plasma sintering), and the properties of the alloys produced by these three methods were compared. The fracture strength of this alloy after spark plasma sintering (1240.5 ± 73.2 MPa) was nearly three times that of vacuum furnace sintering (428.8 ± 42.7 MPa). Investigations by Xu et al. [20] showed that β-Ti has good fatigue resistance. Mechanical alloying, on the other hand, has yielded a variety of results in the creation of titanium-based biomaterials. In addition, spark plasma sintering was employed to create a variety of titanium-based biomaterials. However, when compared to human bone, sintered titanium-based alloys have elastic moduli of about 80–110 GPa, which are higher than that of human bone (4–30 GPa) [7], and thus the mechanical properties still need to be improved.

Via spark plasma sintering (SPS) [21] metal powder is prepressed and filled into a graphite mold of a specific size and specification. Through the electrical conductivity of graphite, a pulse current is applied to it, and the temperature of the metal powder and graphite mold is rapidly increased through the discharge. Simultaneously, during sintering, a particular pressure is applied to the alloy to ensure compaction and to improve the alloy's performance. Spark plasma sintering uses the discharge of pulse currents to convert electric energy into heat, and compared with conventional sintering, it has the advantages of a rapid temperature increase, a short sintering time, a relatively high density of the sintered samples, and a uniform distribution of elements. Furthermore, a sintered sample has a

high purity and a low impurity content as a result of the vacuum sintering process. SPS can quickly produce high-strength and ductile titanium alloy specimens that are suitable for biomedical applications [22,23].

In this paper, a Ti-25Nb-6Zr alloy was prepared using ball milling and spark plasma sintering technology to investigate the effects of different sintering temperatures on the mechanical properties and the elastic modulus of the Ti-25Nb-6Zr alloy, with the goal of determining the best sintering temperature and process conditions. As a result, the alloy's overall mechanical properties were improved, and its elastic modulus was decreased.

## 2. Materials and Methods

### 2.1. Material Preparation

Elemental Ti, Nb, and Zr powders provided by Beijing Goldway Metal Technology Development Co. Ltd. (Beijing, China) with a purity of 99.9% and an average particle size of 50 μm were used as starting materials. The starting powders (80 g) were weighed, and the weight ratio was Ti/Nb/Zr = 6.9:2.5:0.6 in a vacuum glove box. The mixed Ti, Nb, and Zr powders were placed together with stainless-steel balls in a stainless-steel vial for ball milling and mixing with a powder-to-ball weight ratio of 1:10, a rotation speed of 300 rpm/min, and a ball milling time of 1 h under argon atmosphere (Retsch Planetary Ball Mill, PM400, Guangzhou Pufan Scientific Instrument Co., Ltd., Guangzhou, China). Moreover, stainless-steel balls with diameters of 10 mm and 6 mm were used, the weight ratio of which was 3:1, and the stainless-steel vial had an outer diameter of 126 mm, an inner diameter of 90 mm, and a height of 100 mm. Because zirconium powder is combustible, alcohol was added to the milling vial for wet grinding. The weight ratio of alcohol to powder was 6:1. Every 30 min, the ball milling operation was paused for 10 min. The main purpose was to prevent high temperatures during the ball milling process. The milled powder was filtered and dried for 24 h in a vacuum drying oven (DZF-6020, Anhui BEQ Equipment Technology Co. Ltd., Anhui, China). The drying temperature was 323–333 K, and the vacuum degree was $\leq$10 Pa.

### 2.2. Sintering Profiles

The temperature profiles of the sintered Ti-25Nb-6Zr alloy are shown in Figure 1. First, 10 g of the dried powder was filled into a graphite mold with a diameter of 20 mm and a thin graphite foil between the powder and the mold, which was beneficial for removing the sample after sintering, preventing welding, and obtaining a more uniform current flow [24]. Spark plasma sintering (LABOX-650F, Japan sinter land Co., Ltd., Tokyo, Japan) was performed at 1223 K, 1323 K, 1423 K, 1523 K, and 1623 K for a holding time of 10 min. Sintering took place in vacuum at a pressure of 50 MPa (before sintering, the vacuum was manually pressurized to 50 MPa) and a heating rate of 100 K/min. The sintered cylindrical samples with a diameter of 20 mm and a thickness of 5 mm were taken out of the furnace after the heat preservation was completed, and the samples were allowed to cool to room temperature. Finally, the samples were polished to remove the graphite foil from the surface.

### 2.3. Mechanical and Microstructural Characterization

Before characterization, the sample was sufficiently ground and polished to minimize or avoid the effect of carbon on the sample's organization and properties. The Archimedes method using an FK-300Y high-precision multifunctional densitometer (Ka precision measuring instrument Shenzhen Co., Ltd., Shenzhen, China) was employed to determine the density of the sintered specimens. To determine which phases were formed during the sintering process, an X-ray diffractometer (MiniFlex600, Rigaku Corporation, Tokyo, Japan) was utilized to evaluate the phases of the sintered samples. The diffractometer was a Cu target, the operating voltage was 40 KV, the current was 40 mA, the scan rate was 5°/min, the scan angle was 20~80°, and after the test the data were analyzed on MDI Jade software (MDI Jade6.0, Materials Data, Livermore, CA, USA). The XL30ESEM-TMP field emission

scanning electron microscope (FESEM, Nano Nova45, FEI, Eindhoven, The Netherlands) was used to characterize the compressive fracture morphology of the samples for the purpose of analyzing the toughness and brittleness of the specimens. For metallographic observation (BA310Met-T metallurgical microscope, Chengdu Yingdu technology Co., Ltd., Chengdu, China), the samples were polished (the sintered specimens are sanded with sandpaper of grit 120#, 240#, 500#, 800#, 1000#, 2000#, 3000#, and 7000# in sequence and then mechanically polished) and etched with a 1% HF + 3% HNO3 + 96% $H_2O$ solution for 5~8 s. The Vickers hardness was determined using an Mc010 series Vickers hardness tester (Shanghai Yanrun Optical Machine Technology Co., Ltd., Shanghai, China) at a load of 0.1 kgf, a dwell time of 15 s, and a loading speed of 0.05 mm/s. For compression testing, the sintered blocks were cut into small cylindrical specimens of Ø 2.5 mm × 5 mm and polished with sandpaper to remove oil stains and burrs on the surfaces of the specimens. The compressive strength, strain, and elastic modulus were measured using universal testing equipment (AG-X plus, 20KN-50KN, Tokyo, Japan) at a crosshead speed of 0.1 mm/min. The microstructures of the Ti-25Nb-6Zr specimens were revealed by transmission electron microscopy (TEM, Tecnai-G2-TF30-S-Twin, FEI, Amsterdam, The Netherlands).

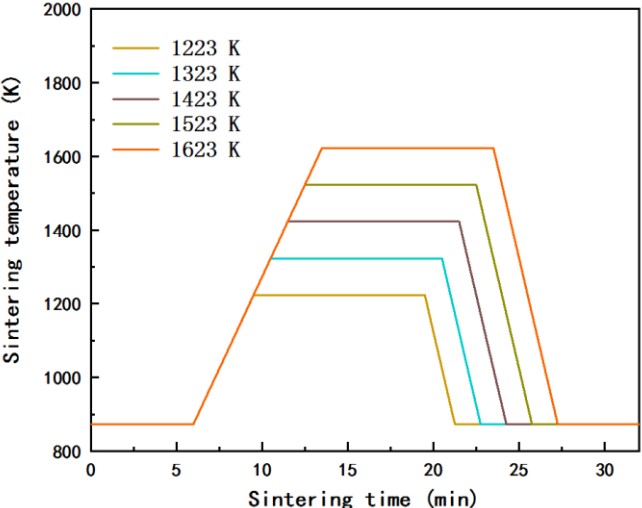

**Figure 1.** Sintering profiles of Ti-25Nb-6Zr alloy.

## 3. Results

The as-received starting materials and the milled product powders had different morphologies and sizes. Figure 2a–c show SEM images of the as-received Ti, Nb, and Zr powders. Ti particles exhibited different irregular shapes, while Nb and Zr exhibited regular spherical shapes. SEM micrographs and EDS mapping of milled Ti-25Nb-6Zr powders are shown in Figure 2d. Compared to the as-received morphology, after milling the average particle size was reduced.

### 3.1. Vickers Hardness and Compressive Properties

The Vickers hardness results of the Ti-25Nb-6Zr specimens sintered at different temperatures are shown in Figure 3 and Table 1, revealing that the Vickers hardness reached its maximum value of 595.505 ± 9.16 HV when the sintering temperature rose to 1623 K. Compared to Ti-6Al-4V (385.48 HV) [24], a regularly used biomedical material, the Vickers hardness of the present material is significantly higher.

The Archimedes method was used to determine the relative density of the sintered samples. The relative density increased from 98.69 ± 0.2% to 99.45 ± 0.2% as the sintering temperature increased from 1223 K to 1623 K (Figure 3 and Table 1). This trend in the density values was correlated with the increasing Vickers hardness. The density and Vickers hardness slightly increased as the temperature increased from 1423 K to 1623 K. A possible explanation for this density and hardness increase may be related to the fact

that plasma was formed between the powder particles during the SPS sintering process using high-frequency discharge, which can establish a sintering neck between two powder particles via element diffusion [25]. As a result, the porosity in the samples shrinks, and the densification process is expedited at a high sintering temperature. Even when the temperature rises, the sample's relative density remains relatively constant.

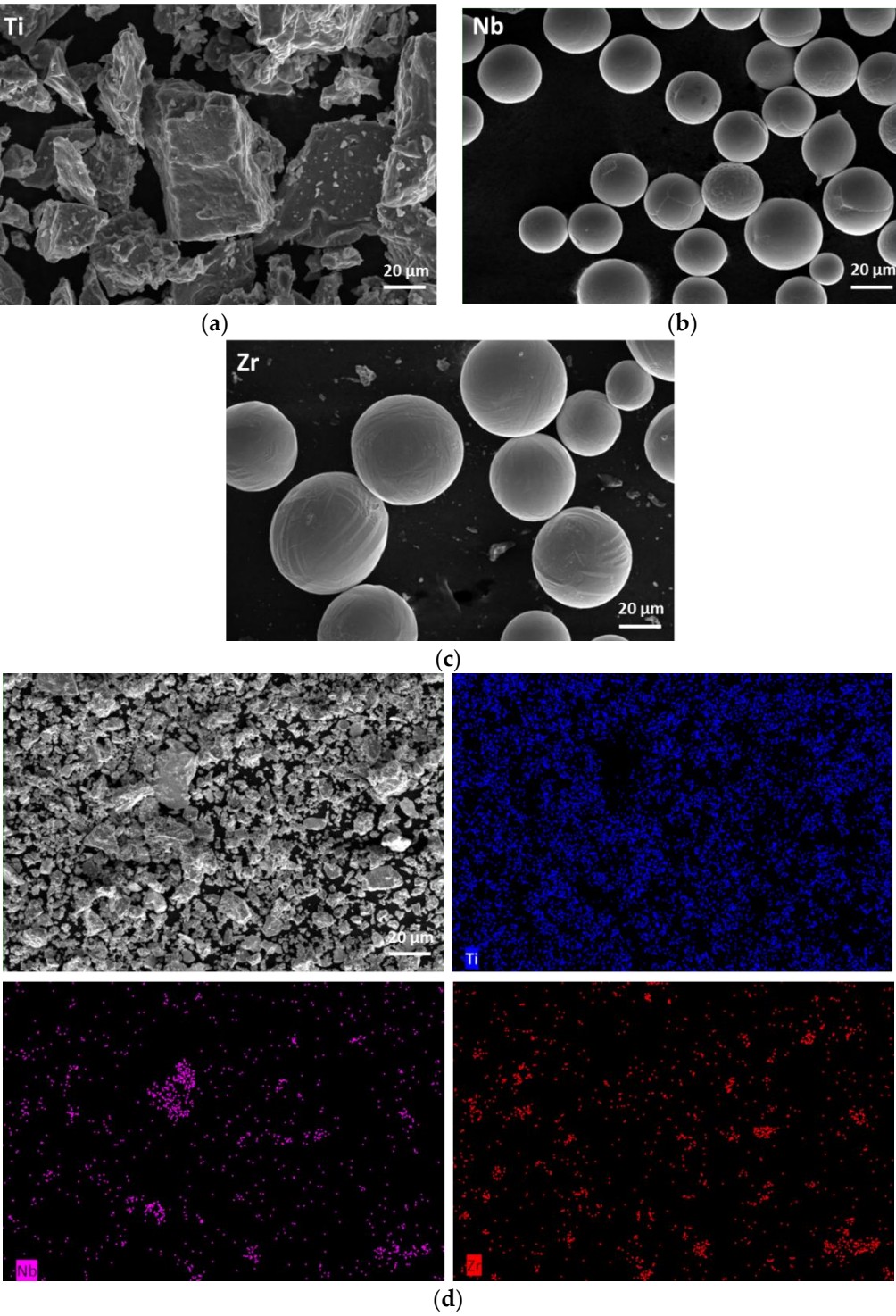

**Figure 2.** SEM micrographs of as-received (**a**) Ti, (**b**) Nb, and (**c**) Zr powders. (**d**) shows milled Ti-25Nb-6Zr powders.

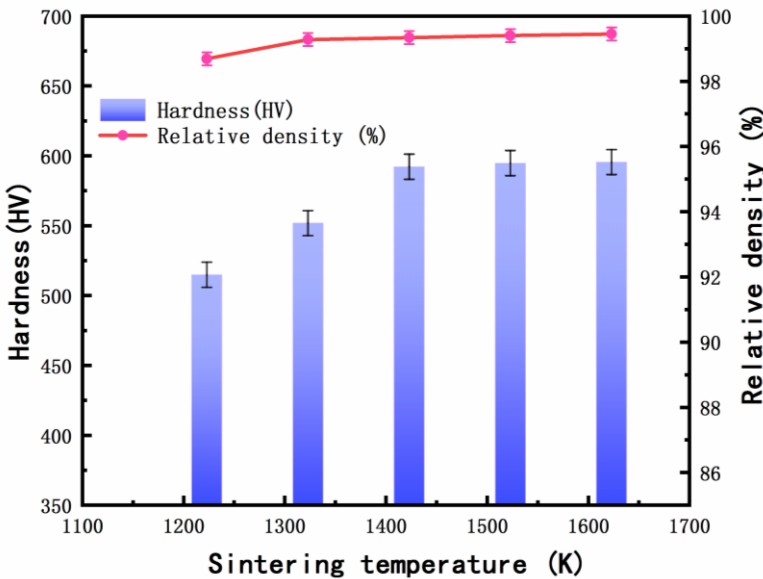

**Figure 3.** Effect of the SPS temperature on the relative density and hardness of Ti-25Nb-6Zr.

**Table 1.** The relative density and hardness of Ti-25Nb-6Zr alloy at different temperatures.

| Temperature (K) | Hardness (HV) | Relative Density (%) |
|:---:|:---:|:---:|
| 1223 | 541.838 ± 8.134 | 98.69 ± 0.2 |
| 1323 | 551.818 ± 9.15 | 99.28 ± 0.19 |
| 1423 | 592.206 ± 8.56 | 99.34 ± 0.15 |
| 1523 | 594.806 ± 9.12 | 99.4 ± 0.2 |
| 1623 | 595.505 ± 9.16 | 99.45 ± 0.2 |

The compressive engineering stress–strain curves of the Ti-25Nb-6Zr alloy sintered at various SPS temperatures are shown in Figure 4 and Table 2. The compressive stress–strain curves of the Ti-25Nb-6Zr alloy were strongly affected by the sintering temperature. As the sintering temperature increased from 1223 K to 1523 K, the ultimate compressive strength of the sample increased from 1445 ± 5 MPa to 1486 ± 5 MPa, and the yield stress of the sample increased from 1340 ± 5 MPa to 1390 ± 5 MPa. The strain, on the other hand, increased from 5.3 ± 0.5% to 7.5 ± 0.5%. For the samples sintered at 1623 K, the compressive strength reached a high value of 1678 ± 5 MPa, with a yield stress of 1495 ± 5 MPa and a strain of 12.4 ± 0.5%. The Ti-25Nb-6Zr samples sintered at low temperatures were not fully densified, resulting in low Vickers hardness and fracture strength. The relative density and Vickers hardness of the Ti-25Nb-6Zr alloy were positively correlated with the sintering temperature. The Ti-25Nb-6Zr alloy had better mechanical properties when sintered at higher temperatures. The variation in the fracture strength of the samples, on the other hand, could be linked to their different microstructures.

A field emission scanning electron microscope was used to examine the compression fracture morphology of samples sintered at different temperatures, as illustrated in Figure 5. As depicted in Figure 5a, the Ti-25Nb-6Zr specimens sintered at 1223 K exhibited a modest amount of quasicleavage and rough dimples on the fracture surface. More dimples could be seen when the sintering temperature rose from 1323 K to 1523 K. A large number of dimples appeared in the fracture morphology of the samples, implying that the ductility increased, in good agreement with Figure 5b–d. When the sintering temperature reached 1623 K, regular dimples in the fracture morphology were visible, revealing that the plasticity of the samples was higher.

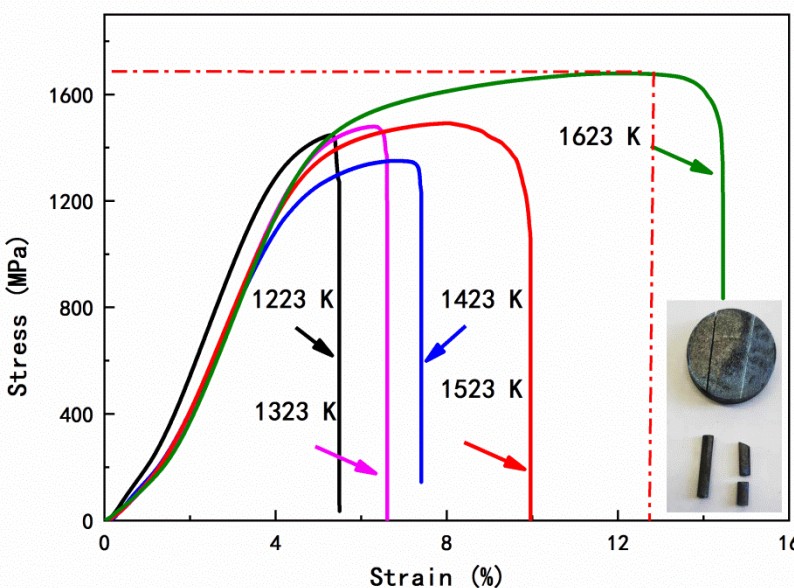

**Figure 4.** Compressive engineering stress–strain curves for samples sintered at different temperatures.

**Table 2.** The compression properties of Ti-25Nb-6Zr alloy at different temperatures.

| Temperature (K) | Ultimate Compressive Strength (MPa) | Yield Stress (MPa) | Strain (%) |
|---|---|---|---|
| 1223 | 1445 ± 5 | 1340 ± 5 | 5.3 ± 0.6 |
| 1323 | 1478 ± 8 | 1407 ± 6 | 6.61 ± 0.4 |
| 1423 | 1349 ± 6 | 1275 ± 5 | 7.41 ± 0.5 |
| 1523 | 1486 ± 5 | 1390 ± 5 | 7.5 ± 0.5 |
| 1623 | 1678 ± 5 | 1495 ± 5 | 12.4 ± 0.5 |

### 3.2. Microstructure and Phase Constitution

The microstructures of the Ti-25Nb-6Zr samples prepared by high-energy ball milling and SPS at different temperatures are presented in Figure 6.

Obviously, all the alloys contained a two-phase region with bcc β-Ti and hcp α-Ti. Meanwhile, the microstructure was of an hcp α-Ti region surrounded by a bcc β-Ti matrix. α-Ti and a small amount of β-Ti in the structure could be observed when the sintering temperatures were 1223 K and 1323 K, as shown in Figure 6a,b. With the increase in the sintering temperature from 1423 K to 1623 K, the grain boundary was clear, and the two regions were visible. The size of the two-phase region increased continuously, as shown in Figure 6c–e.

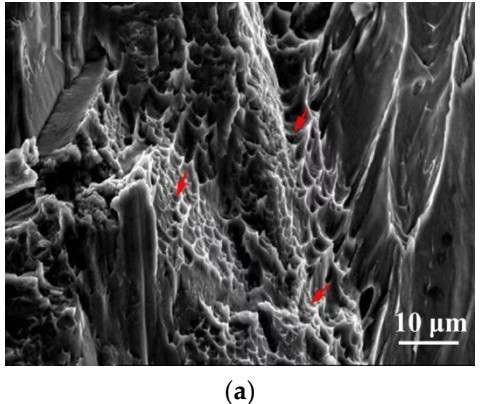

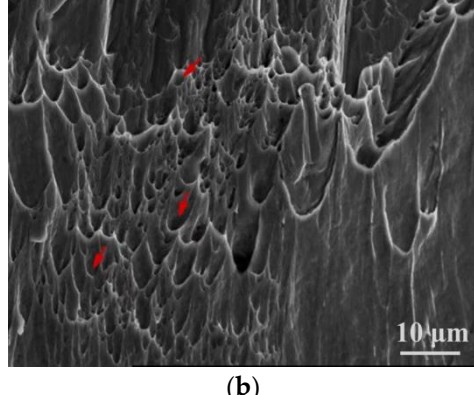

(a)                    (b)

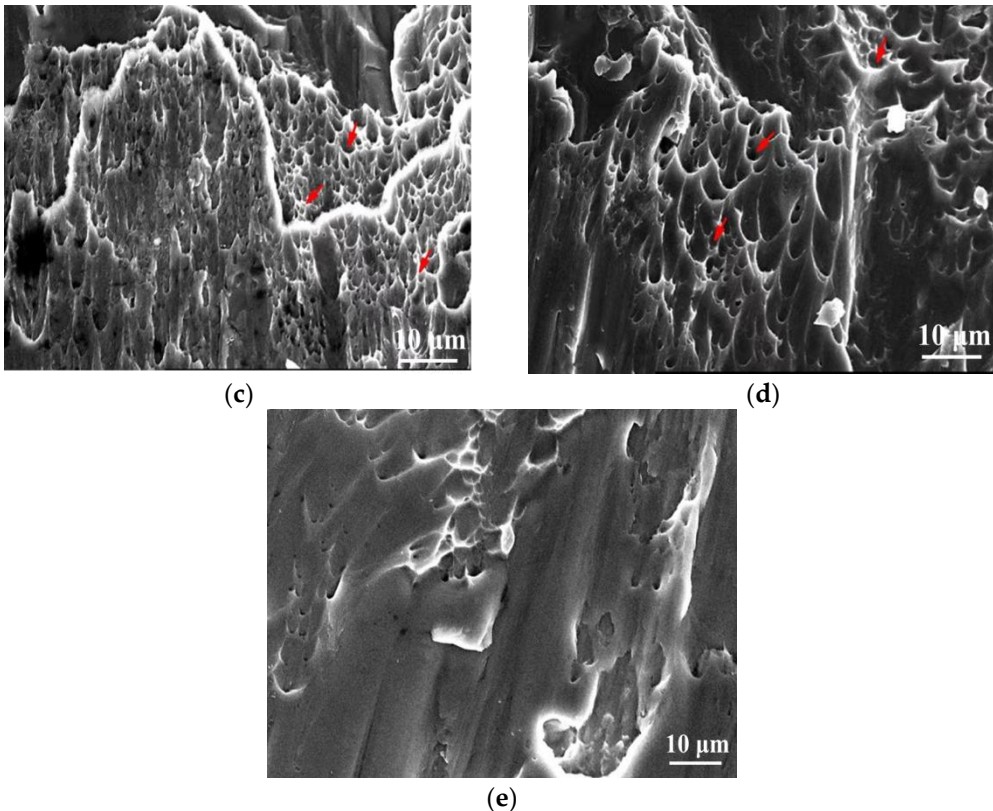

**Figure 5.** Fracture morphology of samples sintered at different temperatures: (**a**) 1223 K, (**b**) 1323 K, (**c**) 1423 K, (**d**) 1523 K, and (**e**) 1623 K.

In order to investigate the elements and phase distribution of the Ti-25Nb-6Zr samples sintered at different temperatures, SEM images and EDS maps were obtained using scanning electron microscopy (SEM) (SEM-EDS), as shown in Figure 7. Apparently, the sintering temperature had a significant effect on the microstructure evolution of the Ti-25Nb-6Zr alloy fabricated by spark plasma sintering. At a low sintering temperature of 1223 K, Nb and Ti elements were unevenly distributed (Figure 7a). This can be explained by the fact that Nb and Ti cannot diffuse efficiently at a low sintering temperature. Figure 7b presents characteristic micrographs of the Ti-25Nb-6Zr alloy sintered at 1323 K, which are consistent with the observed results of the optical microstructure characteristics in Figure 6. It can be seen that Nb and Zr were concentrated and distributed in the black cluster areas in the microstructure, while Ti was the matrix. As revealed in Figure 7c,d, increasing the sintering temperature from 1423 K to 1523 K reduced the black cluster areas, implying a higher level of homogeneous element distribution at higher temperatures than at low temperatures in this area. Nb and Ti were distributed homogeneously after sintering at 1623 K, as shown in Figure 7e, confirming the effective alloying of the powders with increasing sintering temperatures. It can be safely concluded that sintering at a higher temperature promotes element diffusion and the alloying of the original elemental powder mixture, and the element distribution gradually becomes uniform.

The XRD patterns of the Ti-25Nb-6Zr alloy sintered at different temperatures are displayed in Figure 8. It can be seen that the samples sintered at different temperatures were mainly composed of α-Ti, β-Ti, Nb, and a small amount of Zr.

The XRD patterns show that, compared with a pure Ti reference sample sintered at 1523 K, the diffraction peak intensity of α-Ti in the Ti-25Nb-6Zr sintered alloy gradually decreased and the diffraction peaks of β-Ti gradually increased (marked as A and B) when the sintering temperature increased from 1223 K to 1623 K. The β-Ti diffraction peaks became most prominent with sintering at 1623 K. Apparently, higher sintering temperatures promote the formation of β-Ti with a BCC structure. From the three diffraction

peaks of β-Ti, (110), (200), and (211), as shown in Figure 8, the average grain size d with temperature was determined from the peak-broadening analysis using the equation Bcos θ = 0.9λ/d + ηsin θ (where B is the peak width at half maximum intensity, h is the Bragg angel, λ is the wavelength of the X-ray, d is the grain size, and η is the strain) [18]. It was found from Figure 8 that with increasing temperatures the peak broadening decreased, and the grain size, d, increased gradually.

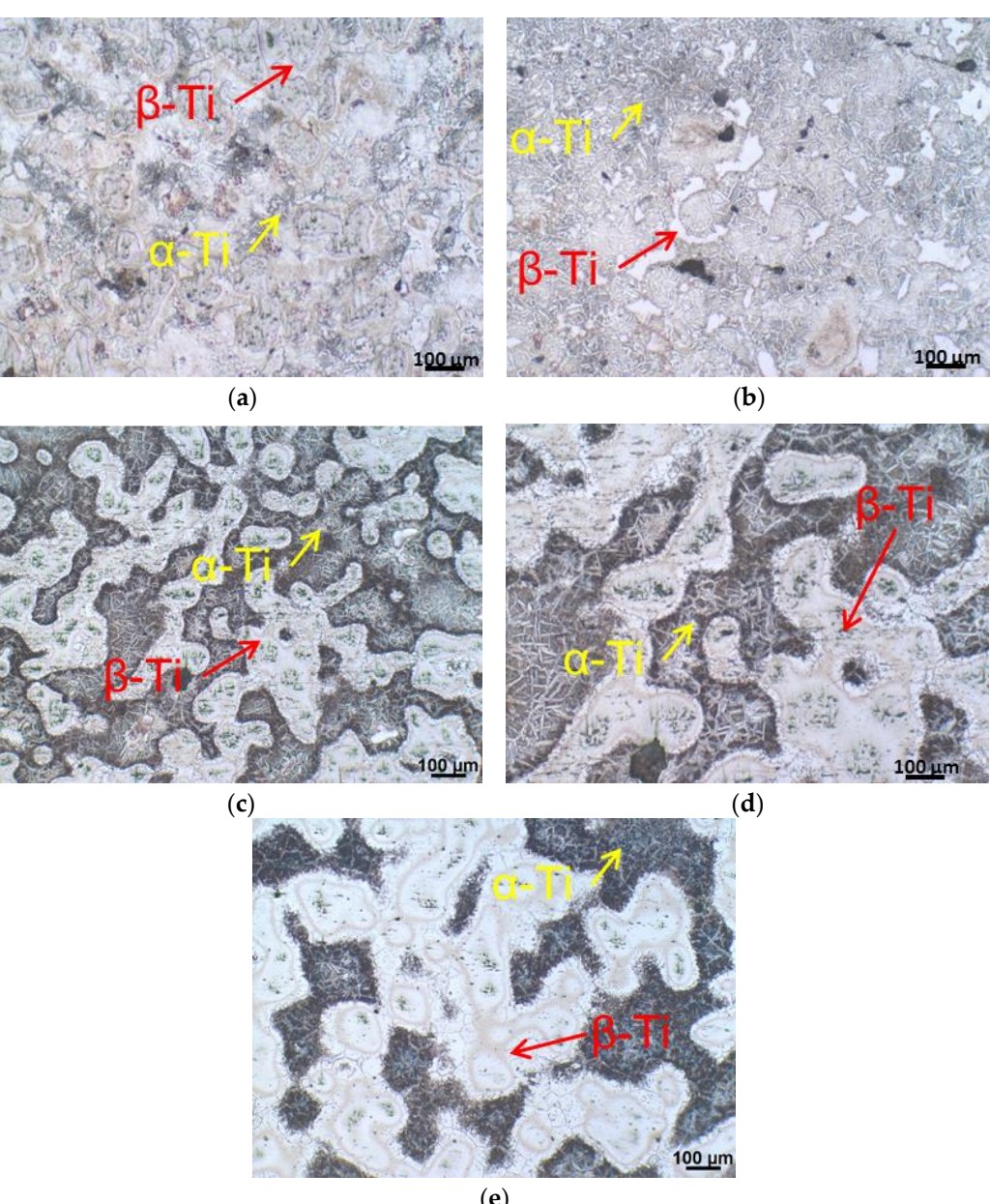

**Figure 6.** Optical micrographs of Ti-25Nb-6Zr specimens sintered at different temperatures: (**a**) 1223 K, (**b**) 1323 K, (**c**) 1423 K, (**d**) 1523 K, and (**e**) 1623 K.

Figure 9a shows a bright-field TEM micrograph of the Ti-25Nb-6Zr alloy sintered at 1623 K. The fast Fourier transform (FFT) in Figure 9b corresponds to the area of grain A in Figure 9a and reveals a BCC-Ti crystal structure. Moreover, it can also be observed that the zone axis of the BCC-Ti structure presented in the pattern is 011. It can be suspected that grain A, with a BCC-Ti crystal structure, is newly formed since α-Ti always has an HCP structure at an ambient temperature. Compared with the α-Ti phase, the β-Ti phase has more slip systems and higher compressive strength. Therefore, the mechanical properties of

the sintered Ti-25Nb-6Zr specimens with higher volume fractions of β-Ti phase produced at higher sintering temperatures were improved, i.e., the compressive strength and the ductility were higher.

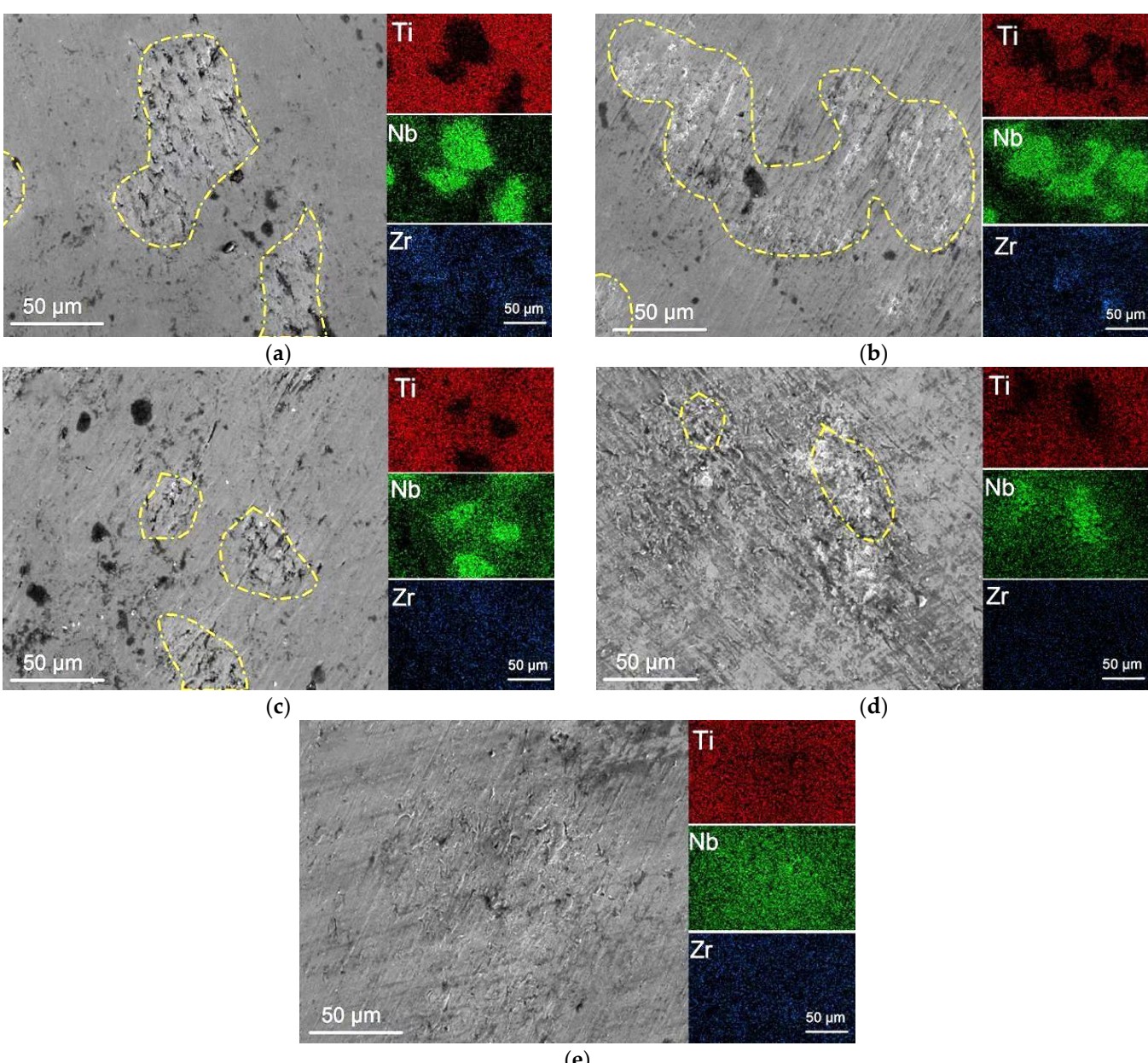

**Figure 7.** SEM images and EDS mapping of Ti-25Nb-6Zr sintered at different temperatures: (**a**) 1223 K, (**b**) 1323 K, (**c**) 1423 K, (**d**) 1523 K, and (**e**) 1623 K.

Combining the XRD patterns, SEM images, and EDS maps of the Ti-25Nb-6Zr specimens sintered at different temperatures, it can be noted that the addition of Nb had a strong effect on the formation of a BCC-Ti crystal structure (as expected since the element is known as a strong β-Ti-forming element [26]). Compared with the pure Ti sintered at 1523 K, the diffraction peaks of β-Ti gradually increased (marked as A and B) when the sintering temperature increased from 1223 K to 1623 K. Nb and Zr could be indefinitely dissolved in Ti to form β-Ti. The diffusion of Nb and Zr in the matrix promoted the formation of β-Ti. The effect of the Nb and Zr contents on the phase formation, microstructure development, and the resulting mechanical properties of such SPS sintered alloys needs to be investigated further.

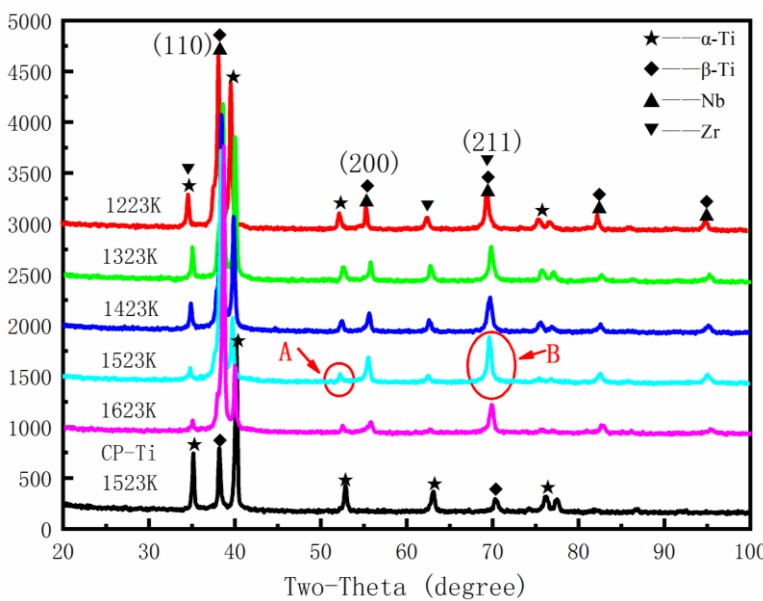

**Figure 8.** XRD patterns of the Ti-25Nb-6Zr alloy after sintering at different temperatures.

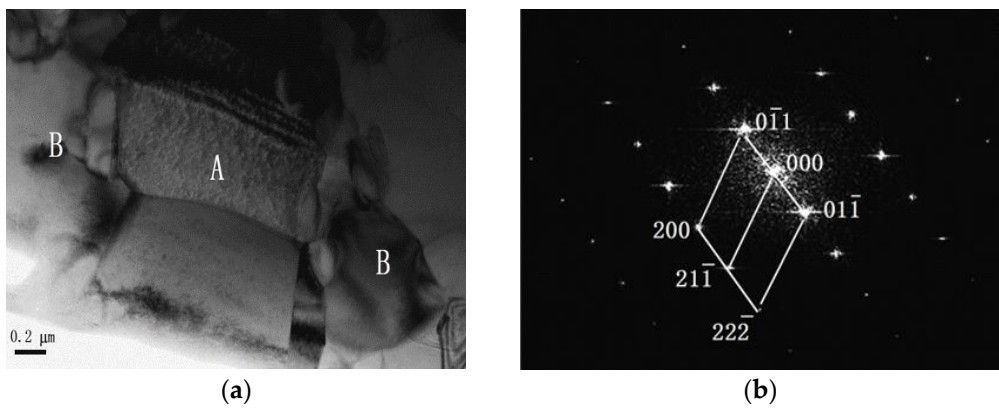

**Figure 9.** (**a**) Bright field TEM micrograph of a Ti-25Nb-6Zr alloy sintered at 1623 K. (**b**) FFT patterns of β-Ti with a BCC structure.

## 4. Conclusions

In this study, high-strength Ti-25Nb-6Zr biomedical alloys with β-Ti structures were prepared by high-energy ball milling and spark plasma sintering (SPS). The effect of different sintering temperatures on the phase formation, microstructure evolution, and mechanical properties of the Ti-25Nb-6Zr alloy were investigated. The results show that the densification process was accelerated with increasing sintering temperatures. Simultaneously, more body-centered cubic titanium (bcc-β-Ti) phase was observed, and the comprehensive mechanical properties were improved. The compressive fracture features revealed a small amount of quasicleavage and dimples, indicating a predominately ductile fracture mode. More dimples were observed in the fracture morphology of the compressed Ti-25Nb-6Zr samples sintered at higher temperatures, indicating that the ductility of the material was enhanced. The mechanical properties of the Ti-25Nb-6Zr biomedical alloy sintered at 1623 K revealed a high compressive strength of 1678 ± 5 MPa and a strain of 14.5 ± 0.5%. The strengthening mechanism was discussed from the aspects of the higher content of bcc-β-Ti phase formed as well as the more homogeneous distribution of Nb and Zr at high sintering temperatures. This work provides a new fabrication method for Ti-25Nb-6Zr biomedical alloy specimens with high strength and low modulus values. A comprehensive study of this technology and its strengthening mechanisms will provide

the basis for the preparation of high-performance Ti-25Nb-6Zr alloy samples with further improved properties.

**Author Contributions:** Conceptualization, F.L.; Methodology, Q.Z.; Software, Q.X.; Validation, K.G.P.; Formal Analysis, Q.Z.; Investigation, Q.Z.; Resources, Q.Z.; Data Curation, P.C.; Writing—Original Draft Preparation, Q.Z.; Writing—Review and Editing, F.L.; Visualization, J.E.; Supervision, J.Y.; Project Administration, F.L.; Funding Acquisition, F.L. All authors have read and agreed to the published version of the manuscript.

**Funding:** This research was funded by the National Science Foundation of China (Project No. 51904133), the Science Foundation of the Yunnan Provincial Science and Technology Department (No. 202101AT070085 and 202001AT070082), and the Science and Technology Major Project of Yunnan Province (No. 202202AG050004).

**Data Availability Statement:** The data presented in this study are available on request from the corresponding author.

**Conflicts of Interest:** The authors declare no conflict of interest. The funders had no role in the design of the study; in the collection, analyses, or interpretation of the data; in the writing of the manuscript; or in the decision to publish the results.

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
