# Peer review of "Mechanical Properties and Microstructural Evolution of Ti-25Nb-6Zr Alloy Fabricated by Spark Plasma Sintering at Different Temperatures"

_metals, doi:10.3390/met12111824_

Round 1

Reviewer 1 Report (New Reviewer)

This paper describes an interesting approach for the fabrication of a biomedical alloy. The paper is well written! Very clear and understandable in all parts. There is a clear line from the introduction, the experimental details to the results and their discussion. The conclusions are convincing.   

Author Response

Dear reviewers,

      We thank the your highly positive comments to our paper. When writing this article, we did a lot of preliminary preparation and research, and the experiment was also carried out according to scientific methods. We several authors reached the final result after repeated discussion. Thank you again for your recognition of this paper!

Yours sincerely,

FENGXIAN LI

Reviewer 2 Report (New Reviewer)

The paper is interesting but needs mandatory revisions and complementary analyses prior to publication.

Line 38:  Mpa should be replaced by MPa

Is there any grain size modification after the milling step
Is there any contamination of the powder coming from the milling tools (balls & vial)

After SPS, is there any carbon contamination coming from the gaphite (foil & molds), if yes to what extent.

Authors mention that:" A possible explanation for this density and hardness increase may be related to the fact that plasma was formed between the powder particles during the SPS sintering process using high-frequency discharge". They have to argue on that - what are the signatures that allow them to confirm the presence of a plasma?

Authors mention that: "Furthermore, plastic deformation of particles happens under the influence of axial pressure, temperature field, and voltage field". They have to argue on that, what are the data, characterizations, which confirm that the plastic deformation of the particles is responsible for the densification of the material.  What allows them to eliminate any other mechanism?

How do they explain that densification decreases for the highest temperature (i.e. 1623K)?

For the readability of the article, authors should put all the data (density, hardness, ultimate compressive strength, yield stress and strain) in a single table.

Authors mention that: "The variation in fracture strength of the samples, on the other hand, can be linked to their different microstructures."  The characterizations of the microstructures of the different samples should be deeply improved to argue this. Is there any change in the grain sizes with the sintering temperature?

Authors mention that: "With the increase of the sintering temperature from 1523 K to 1623 K, the microstructure becomes more homogeneous, as shown in Figure. 6(d) and (e).". There are no significant differences between images (b & d) of Figure 6. Did they quantify this by image analysis?

Authors should analyze more in details the XRD patterns in using Rietveld method in order to follow with the sintering temperature the evolution of the different phases (cell parameters to see any alloying effect) and to quantify them.

Authors mention that: "The absence of Nb and Zr in the matrix promotes the formation of β-Ti, which can further improve the mechanical properties of sintered Ti-25Nb-6Zr alloy". This sentence should be re written as Nb and Zr are still present in the materials.

Authors should discuss more in details the increase of the mechanical properties (ultimate compressive strength, yield stress and strain) with the presence of β-Ti.

Finally, to explain the decrease in density for the higher sintering temperature can we evoke a potential Kirkendall effect with the alloy formation?

Author Response

Response to Reviewer 2 Comments

Thank you very much for your kindly comments on our manuscript entitled “Mechanical properties and microstructural evolution of Ti-25Nb-6Zr alloy fabricated by spark plasma sintering at dif-ferent temperatures” (ID: metals-1921828). Those comments are all valuable and very helpful for revising and improving our manuscript. We have considered reviewer’s comments carefully and have made corrections which we hope to meet with your approval. Please see our point to point responses to all your comments below. The corresponding revisions in the body of manuscript are marked in red.

Thank you again for your consideration.

Point 1: Line 38:  Mpa should be replaced by MPa.

Response 1: Thank you very much for your good suggestions.We have revised it.

Point 2: Is there any grain size modification after the milling step.

Response 2: We appreciate the reviewer’s suggestion. We have added the SEM images of the powder before and after ball milling to the manuscript and we found that the shape of the powder changed before and after ball milling, shown in Figure. 2.

Point 3: Is there any contamination of the powder coming from the milling tools (balls & vial).

Response 3: Thank you very much for your question, we are very happy to answer it for you. We do our best to avoid contamination of the metal powder before and after ball milling. Before and after ball milling, we put the small balls into the ball mill jar and clean them well with alcohol on the ball mill, then dry and weigh the mass of the small balls, there is almost no change, so we think the small balls and the ball mill jar have almost no contamination to the powder. 

Point 4: After SPS, is there any carbon contamination coming from the gaphite (foil & molds), if yes to what extent.

Response 4: We appreciate the reviewer’s suggestion.After SPS, we polished the whole sample surface sufficiently with sandpaper, and no carbon contamination was found in the later measurements, so we think there is almost no carbon contamination after sintering.

Point 5: Authors mention that:" A possible explanation for this density and hardness increase may be related to the fact that plasma was formed between the powder particles during the SPS sintering process using high-frequency discharge". They have to argue on that - what are the signatures that allow them to confirm the presence of a plasma?

Response 5: Thank you very much for your valuable comments, for this problem we are citing the conclusions obtained from the study of Z.H. Zhang et al. (Reference 25)to explain, while forming the sintered neck without the relevant tests.

Point 6: Authors mention that: "Furthermore, plastic deformation of particles happens under the influence of axial pressure, temperature field, and voltage field". They have to argue on that, what are the data, characterizations, which confirm that the plastic deformation of the particles is responsible for the densification of the material.What allows them to eliminate any other mechanism?

Response 6: Thank you very much for your valuable comments. We removed it.

Point 7: How do they explain that densification decreases for the highest temperature (i.e. 1623K)?

Response 7: Thank you very much for your comments. We can see that the densities and hardnesses do not change much when the temperature increases to 1423 K. Since the readings are given with errors, we consider that the densities and hardnesses are within the error range and can be considered unchanged. Not only that, we have re-measured the relative density and hardness as shown in Fig. 3 and also corrected the less appropriate description in the manuscript.

Point 8: For the readability of the article, authors should put all the data (density, hardness, ultimate compressive strength, yield stress and strain) in a single table.

Response 8: Thanks for your kindly comments on our manuscript. We have organized the data into tables, shown in Table 1 and Table 2.

Point 9: Authors mention that: "The variation in fracture strength of the samples, on the other hand, can be linked to their different microstructures."  The characterizations of the microstructures of the different samples should be deeply improved to argue this. Is there any change in the grain sizes with the sintering temperature?

Response 9: Thank you very much for your valuable comments, we have re-corroded the metallographic images and modified the corresponding places in the manuscript as shown in Figure 6.

Point 10: Authors mention that: "With the increase of the sintering temperature from 1523 K to 1623 K, the microstructure becomes more homogeneous, as shown in Figure. 6(d) and (e).". There are no significant differences between images (b & d) of Figure 6. Did they quantify this by image analysis?

Response 10: Thank you very much for your valuable comments. From the organization obtained by our re-etching, it can be obtained that the grain size gradually increases with the increase of temperature.

Point 11: Authors should analyze more in details the XRD patterns in using Rietveld method in order to follow with the sintering temperature the evolution of the different phases (cell parameters to see any alloying effect) and to quantify them.

Response 11: We thank the reviewer’s suggestion. We try our best to analyze the XRD as follows: From three diffraction peaks of β-Ti (110), (200) and(211), as shown in Figure 8, the average grain size d with temperature was determined from the peak broadening analysis using the equation Bcos θ=0.9λ/d+ηsin θ(where B is the peak width at half maximum intensity,h is the Bragg angel, λis the wavelength of the X-ray, d isthe grain size and η is the strain )[18]. It is found from Figure.8 that with increasing temperatures, the peak broadening decreases ,the grain size d increases gradually.

Point 12: Authors mention that: "The absence of Nb and Zr in the matrix promotes the formation of β-Ti, which can further improve the mechanical properties of sintered Ti-25Nb-6Zr alloy". This sentence should be re written as Nb and Zr are still present in the materials

Response 12: Thank you very much for your valuable comments. It is correct.We have rewritten the sentence as: Nb and Zr can be indefinitely dissolved in Ti to form β-Ti ,the diffuse of Nb and Zr in the matrix promotes the formation of β-Ti.

Point 13: Authors should discuss more in details the increase of the mechanical properties (ultimate compressive strength, yield stress and strain) with the presence of β-Ti.

Response 13: We appreciate theyour’s suggestion. We removed it.

Point 14: Finally, to explain the decrease in density for the higher sintering temperature can we evoke a potential Kirkendall effect with the alloy formation?

Response 14: We appreciate theyour’s suggestion. Maybe potential Kirkendall effect exists, but we haven't further analyzed it. However, we re-measure the density and hardness, within the error range, and regard them as almost unchanged.

Besides the above changes, we also corrected some expression errors. We appreciate for Editors/Reviewers’ comments and suggestions earnestly, and hope that the corrections will meet with your approval.

From your comments, it’s obvious that you are seasoned in these aspects and it’s my great honor to obtain your good advices. Thank you for your suggestions.

Thank you again for your time and consideration.

We look forward to your reply to our revised manuscript.

Yours sincerely,

FENGXIAN LI

Reviewer 3 Report (New Reviewer)

Dear authors

your work is interesting an can be published after some necesary improovements.

The schematics in fig 2 is correct however it brings nothing to this research. I suggest removing it

Please check your crosshead speed of (0.1 mm/min ) it can be high for such a small sample height and the test may not be considered as done in a quasi static regime.

How many tests were done for each sample? Is a such small sample relevant? 

The optical images in fig 6 presents no microstructure. different etchant may reveal the microstructure, so the corresponding discussion is pointless.

Please recheck the conclusions paragraph the folowing paragraph is not supported by the data "In this study, high-strength Ti-25Nb-6Zr biomedical alloys with β-Ti structure were prepared by high-energy ball milling ...." in my oppinion there was no milling. Homogenisatiom yes but no milling; if the authors would have conducted mechanical milling  most likely the powders in their elemental form would not be present .

Author Response

   Thank you very much for your kindly comments on our manuscript entitled “Mechanical properties and microstructural evolution of Ti-25Nb-6Zr alloy fabricated by spark plasma sintering at dif-ferent temperatures” (ID: metals-1921828). Those comments are all valuable and very helpful for revising and improving our manuscript. We have considered reviewer’s comments carefully and have made corrections which we hope to meet with your approval. Please see our point to point responses to all your comments below. The corresponding revisions in the body of manuscript are marked in red.

Thank you again for your consideration.

Point 1: your work is interesting an can be published after some necesary improovements.

Response 1: We appreciate the referee’s highly positive comments to our paper. We have revised the manuscript carefully according to your comments.

Point 2: The schematics in fig 2 is correct however it brings nothing to this research. I suggest removing it

Response 2: We appreciate the reviewer’s suggestion. We removed it.

Point 3: Please check your crosshead speed of (0.1 mm/min ) it can be high for such a small sample height and the test may not be considered as done in a quasi static regime.

Response 3: We appreciate the reviewer’s suggestion. By consulting the literature, we found that the strain rate used by Shengping Si et al. in compression test is 2.4×10-3 s-1, and the size of compression pattern is∅4×4 mm. The mechanical properties of the specimens of 3 mm in diameter and 6 mm in length under compression were tested in a universal testing machine at a strain rate of 5 × 10-3 s-1 by H.Z. Lu et al.What’s more, the compression properties were tested at a strain rate of 8×10-3s-1, the specimens of 3 mm in diameter and 6 mm in length . Our sample size is small, and the crosshead speed is also low. Therefore, we think this test can be regarded as being carried out under quasi-static pressure.

Point 4: How many tests were done for each sample? Is a such small sample relevant?

Response 4: We appreciate the reviewer’s suggestion.After sintering, the sample was first polished and cleaned, and tested for density, hardness, and XRD in turn without damaging the sample, then three cylinders (Ø 2.5 mm × 5 mm) were cut for compression testing, and the remaining sample was polished and tested for metallography and SEM, and the sample was sufficient for testing.

Point 5: The optical images in fig 6 presents no microstructure. different etchant may reveal the microstructure, so the corresponding discussion is pointless.

Response 5: Thank you very much for your valuable comments, we have re-corroded the metallographic images and modified the corresponding places in the manuscript as shown in Figure 6.

Point 6: Please recheck the conclusions paragraph the folowing paragraph is not supported by the data "In this study, high-strength Ti-25Nb-6Zr biomedical alloys with β-Ti structure were prepared by high-energy ball milling ...." in my oppinion there was no milling. Homogenisatiom yes but no milling; if the authors would have conducted mechanical milling  most likely the powders in their elemental form would not be present .

Response 6: Thank you very much for your comments. We have added the SEM images of the powder before and after ball milling to the manuscript and we found that the shape of the powder changed before and after ball milling, shown in Figure. 2.

Besides the above changes, we also corrected some expression errors. We appreciate for Editors/Reviewers’ comments and suggestions earnestly, and hope that the corrections will meet with your approval.

From your comments, it’s obvious that you are seasoned in these aspects and it’s my great honor to obtain your good advices. Thank you for your suggestions.

Thank you again for your time and consideration.

  We look forward to your reply to our revised manuscript.

Yours sincerely,

FENGXIAN LI

Round 2

Reviewer 2 Report (New Reviewer)

The authors took into account most of the referee’s comments and suggestions. The paper can be published

Reviewer 3 Report (New Reviewer)

I can accept your responses and improvements.

This manuscript is a resubmission of an earlier submission. The following is a list of the peer review reports and author responses from that submission.

Round 1

Reviewer 1 Report

Review of the article entitled

 Effect of sintering temperature on microstructure evolution and 2 mechanical properties of Ti-25Nb-6Zr fabricated by spark 3 plasma sintering

The article deals with the elaboration of Ti-Nb-Zr alloys by mechanical alloying and spark plasma sintering. The article is well written and the English is good, as far as I can consider, being a non-native English speaker. 

There are 29 remarks on the article inserted as comments in the pdf attached here. 

The article is interesting and the work is well made. However, there are some flaws in the article that must be corrected prior to publication. These corrections will improve the quality of the article. 

The main comments are some missing informations about how the strain was measured and thus how the Young modulus was determined. Is is unacceptable that the value of the Young modulus is presented for the first time in the conclusion part and there is no trace of it in the Results part. 

Also, there are some missing elements in the Materials and methods section. Please refer to the comments in the pdf attached.

The SPS sintering was made using graphite sheets to avoid sticking to the graphite mold. Thus a careful attention must be devoted to the carbon contaminations. The carbides must be carefully considered in the XRD patterns and possible carbon local enrichments must be investigated. The simplest mode is to evaluate the carbon composition in the dark spots observed on the samples surface after polishing. Even if the carbon quantification is difficult by ESD, it can be however possible to see if there is carbon reinforced presence in the dark spots. One can see that the dark spots are Nb-enriched. EDS spectra and quantification would be interesting to be given to show the level of this enrichment. 

The XRD section is not presented in the best way. Thus, the XRD patterns are illisibles and looking for eventual carbides is impossible. simple superposition of line spectra, with eventual y-axis offset is  way better solution.

There are some wrong informations about the Zr effect on the beta phase of Ti. Thus, the authors claim that the Zr is a beta stabilizing element which is not true. With Sn, Zr is the most neutral element concerning the phase stabilization of Ti. Oppositely, the carbon (which was not investigated) is a notorious alpha-stabilizing element.

Also, there is no discussion section, which is not really a problem if there is a paragraph in the Results (which must become Results and Discussion) section. The last paragraph of the Results section must be reinforced.

For all this reasons I suggests a major revision of this paper, in order to improve its quality.

Author Response

Response to Reviewer 1 Comments

Point 1: The article deals with the elaboration of Ti-Nb-Zr alloys by mechanical alloying and spark plasma sintering. The article is well written and the English is good, as far as I can consider, being a non-native English speaker

Response 1: Thank you very much for your affirmation. I still have many shortcomings. I will continue to learn to make my English better in the follow-up study and life, and thank you very much for your sincere suggestions on my manuscript.

Point 2: There are 29 remarks on the article inserted as comments in the pdf attached here.

Response 2:

Line 21: Your suggestions are very useful to us. We have explained it in the manuscript.

Line 77-78: Thank you very much for your good suggestions .In reference 18, the scholars used mechanical alloying, vacuum furnace sintering, and spark plasma sintering to make Ti-26Nb-5Ag alloy and compared them.

Line 111-115: Thank you very much for your good suggestions.We marked in the manuscript.The metal jar has an outer diameter of 126mm, an inner diameter of 90mm and a height of 100mm.

Line 116-117: Thank you very much for your good suggestions.The weight ratio of alcohol to powder is 6:1. We have perfected it in the manuscript.

Line 118: As you know.Every 30 minutes, the ball milling operation is suspended for 10 minutes in order to dissipate heat.

Line 124: Thank you very much for your good suggestions. We manually pressurized it to 50MPa before sintering.

Line 147: The compressed sample is cut by wire cutting machine, and the oil stain and burr on the surface can't be cleaned by ultrasonic, so it needs to be polished with sandpaper.

Line 149: Thank you very much for reminding we.We revised it in the manuscript.

Line 153: Thank you very much for your good suggestions.Sorry, we didn't save the sintering curve data, Only part of the screenshots are saved, so they are not placed in the manuscript, as shown in the following figure:

Line 161: The Archimedes method using a FK-300Y high-precision multifunctional densitom-eter was employed to determine the density of the sintered specimens. 

Line 164-168: Thank you very much for your good suggestions. For the description of the sintering neck, we have some problems. During the SPS sintering process using high frequency discharge, a plasma is formed between the powder particles, which can establish a sintering neck between two powder particles by elemental diffusion and solidification, which is the reason for the increase in density. The evaporation of the elements, may be due to the fact that too high a temperature can cause overburning. We have therefore made some changes in the manuscript, as follows: A possible explanation for this density and hardness increase may be related to the fact that plasma was formed between the powder particles during the SPS sintering process using high-frequency discharge, which can establish a sintering neck between two powder particles via element diffusion and solidification.

Line 176: Thank you very much for your good suggestions.I'm sorry for misleading you. On the one hand, it's the light problem when taking pictures, and on the other hand, it's because the compressed samples don't have very high requirements on the surface. We simply cleaned the oil stains and polished the burrs, so the surface of the samples was dark and didn't show silver.

Line 178: Thank you very much for your good suggestions.The compressive strength, strain and elastic modulus were measured using a universal testing equipmen,and the stress and strain data in the whole compression process are recorded directly by computer.

Line 190: We are very sorry that we didn't explain it clearly when writing the paper.Young's modulus is obtained from the report of compression test.

Line 222-225: Your suggestion is very useful.Although it is not accurate to use XRD and EDS to detect carbon content, the existence of carbon element is not found in our measured XRD and EDS data, which we have not discussed in this article.

Line 228: Thank you very much for your good suggestions.As for your question, we have carried out EDS quantitative analysis on this part before, but because of the low carbon content, we didn't find carbon in EDS analysis, which may be the reason why EDS is not very accurate in quantitative analysis of elements.

Line 235-236: Thank you very much for your kind reminder. After carefully polishing the surface of the sample, we found no carbide formation of Nb or Zr in XRD analysis, so we did not analyze and discuss carbon, and the detection of carbon element was indeed lacking.

Line 238-239: Thank you very much for your advice. We have polished our samples very carefully before testing, as follows: The sintered specimens are sanded with sandpaper of grit 120#, 240#, 500#, 800#, 1000#, 2000#, 3000#, 7000# in sequence, and then mechanically polished. And I have added to the manuscript in the appropriate places.

Line 255: Thank you very much, we have redrawn the XRD pattern. As for the detection of carbon, we have taken it into account when we drew the XRD pattern, but because of the low content of carbon (less than 5%), we didn't find the diffraction peak of carbon in the XRD data.

Line 282: Thank you very much for your good suggestions. We quite agree with your point of view that Zr is not a stable element ofβ-Ti, and Zr contributes a lot to the hardness of this alloy, which is also mentioned in many literatures, (for example: The effect of alloying elements on densification and mechanical behaviour of titanium based alloy.) so we made some modifications in the manuscript.

Point 3: The article is interesting and the work is well made. However, there are some flaws in the article that must be corrected prior to publication. These corrections will improve the quality of the article.

Response 3: Thanks very much for reading our article carefully and making comments. We have carefully consulted your opinions and made changes to try our best to avoid making mistakes again. There are still many shortcomings, and I hope you can understand them.

Point 4: The main comments are some missing informations about how the strain was measured and thus how the Young modulus was determined. Is is unacceptable that the value of the Young modulus is presented for the first time in the conclusion part and there is no trace of it in the Results part.

Response 4: We are very sorry that this is our mistake, we have answered each of the questions in the PDF you marked.

Point 5: Also, there are some missing elements in the Materials and methods section. Please refer to the comments in the pdf attached

Response 5: Thank you very much for your suggestion, we have tried our best to improve the manuscript with some missing contents in the materials and methods section, as shown in the second part of the response to the annotated PDF.

Point 6: The SPS sintering was made using graphite sheets to avoid sticking to the graphite mold. Thus a careful attention must be devoted to the carbon contaminations. The carbides must be carefully considered in the XRD patterns and possible carbon local enrichments must be investigated. The simplest mode is to evaluate the carbon composition in the dark spots observed on the samples surface after polishing. Even if the carbon quantification is difficult by ESD, it can be however possible to see if there is carbon reinforced presence in the dark spots. One can see that the dark spots are Nb-enriched. EDS spectra and quantification would be interesting to be given to show the level of this enrichment.

Response 6: Thank you very much for your sincere advice, you are right, we do use graphite flakes in SPS sintering for this reason, we polished the samples carefully before characterization to minimize carbon contamination, so we did not find carbon in XRD as well as EDS data, of course this could be the reason for the low carbon content, not only that, we also know that EDS in quantitative analysis has certain inaccuracy. We also examined the black aggregates in the sample and found them to be enriched in Nb, again no carbon was measured, for which we are sorry that we are not performing carbon analysis.

Point 7: The XRD section is not presented in the best way. Thus, the XRD patterns are illisibles and looking for eventual carbides is impossible. simple superposition of line spectra, with eventual y-axis offset is  way better solution.

Response 7: Thank you very much for your suggestion, we have modified the XRD pattern.

Point 8: There are some wrong informations about the Zr effect on the beta phase of Ti. Thus, the authors claim that the Zr is a beta stabilizing element which is not true. With Sn, Zr is the most neutral element concerning the phase stabilization of Ti. Oppositely, the carbon (which was not investigated) is a notorious alpha-stabilizing element.

Response 8: Thank you very much for your suggestion, we agree with your idea, which has been revised where relevant in the manuscript.

Point 9: Also, there is no discussion section, which is not really a problem if there is a paragraph in the Results (which must become Results and Discussion) section. The last paragraph of the Results section must be reinforced.

Response9: Thank you very much for your suggestion, we agree with your idea, which has been revised where relevant in the manuscript.

Point 10: For all this reasons I suggests a major revision of this paper, in order to improve its quality

Response10: Thank you very much for your good suggestion. We habitually use this way of writing. In addition, according to your comments, we have revised the article with error.

Article File with Track CHanges,Please see the attachment.

From your comments, it’s obvious that you are an expert. Moreover, we revised some sentences in order to make the manuscript more concise. We hope that the language is now acceptable for publication. Thank you again for your time and consideration.

We appreciate for Editors/Reviewers’ comments and suggestions earnestly. we corrected some expression errors, and hope that the corrections will meet with your approval.

Thank you again for your time and consideration.

    We look forward to your reply to our revised manuscript.

Yours sincerely,

FENGXIAN LI

Reviewer 2 Report

The authors of the manuscript " Effect of sintering temperature on microstructure evolution and mechanical properties of Ti-25Nb-6Zr fabricated by spark plasma sintering" have carried out an investigation to obtain and study a low-modulus Ti-25Nb-6Zr biomedical alloy with a β structure. The microstructure and mechanical properties of the obtained materials were examined as a function of the sintering temperatures.

My comments are the following:

1 – in lines 80 and 86 correct the units of the modulus of elasticity and fracture.

2 – I consider that the text of section 2.1 should go in the introduction.

3 – In section 2.1 should be a detailed description of the raw materials, the analysis equipment used and the conditions of their work during measurement and sintering.

4 – line 112, what are the measurements of the metal jar?

5 - What is the influence of the jar and the balls material in the final alloy? It means the contamination.

6 – Is the heating rate 373K/min correct? or did you mean 100 C/min? If the value 373K/min is correct, give an explanation of why you chose this value.

7 – Give a a detailed explanation of the fracture stress measurement process.

8 – after line 159 figure 2 should be.

9 – Please, provide images that justify the following: “A possible explanation for this density and hardness drop may be related to the fact that plasma was formed between the powder particles during the SPS sintering process using high-frequency discharge, which can establish a sintering neck between two powder particles via element evaporation and solidification.”

10- What the image at the bottom right of figue 3 shows?

11 – Images 4a and b are the same, correct it. And correct the text.

12 – There is no explanation of image 5c, add it.

13- The Zr spectrum in image 6 is not understandable, correct it.

14 – Include a graph with the evolution of grain size according to temperature.

15 – Include a graph with the sintering process, where the heating process, force application and piston displacement depending on temperature are shown.

16 – The English must be corrected.

17- References 12 and 24 are the same.

Author Response

Response to Reviewer 2 Comments

Point 1: in lines 80 and 86 correct the units of the modulus of elasticity and fracture. 

Response 1: Thank you very much for your good suggestions. According to your suggestion, we revised the manuscript carefully and tried to minimize typographical and spelling error.)

Point 2: I consider that the text of section 2.1 should go in the introduction.

Response 2: Thank you very much for your good suggestions. We think your suggestion is very good, but after we have thought about it, we feel that it is necessary to highlight our production process in this section, so it is better to write it as a separate section.

Point 3: In section 2.1 should be a detailed description of the raw materials, the analysis equipment used and the conditions of their work during measurement and sintering.

Response 3: Thank you very much for your good suggestions. We have supplemented this part and marked it in the red part of the article.

Point 4: line 112, what are the measurements of the metal jar?

Response 4: Thank you very much for your good suggestions. For this part, we feel that the size of the ball mill tank is determined by the specifications of the ball mill and has little effect on ball milling, therefore, we do not make a description.The metal jar has an outer diameter of 126mm, an inner diameter of 90mm and a height of 100mm.

Point 5: What is the influence of the jar and the balls material in the final alloy? It means the contamination.

Response 5: Thank you very much for your good suggestions. We try our best to avoid material contamination during ball milling. Before ball milling, we add alcohol to the ball milling tank to clean it for 8 hours, then add titanium powder to ball mill for 8 hours, and the pellets will be wrapped with titanium powder to reduce pollution.

Point 6: What is the influence of the jar and the balls material in the final alloy? It means the contamination.

Response 6: Thank you very much for your good suggestions. 373K/min means 100℃/min,T=t℃+273.15K≈t℃+273K.

Point 7: Give a a detailed explanation of the fracture stress measurement process.

Response 7: Thank you very much for your good suggestions.Fracture stress is obtained by compression test on universal mechanical testing machine, and the stress and strain data in the whole compression process are recorded directly by computer.

Point 8: after line 159 figure 2 should be.

Response 8: Thank you very much for your good suggestions. Have been adjusted.

Point 9: Please, provide images that justify the following: “A possible explanation for this density and hardness drop may be related to the fact that plasma was formed between the powder particles during the SPS sintering process using high-frequency discharge, which can establish a sintering neck between two powder particles via element evaporation and solidification.”

Response 9: Thank you very much for your good suggestions. For the description of the sintering neck, we have some problems. During the SPS sintering process using high frequency discharge, a plasma is formed between the powder particles, which can establish a sintering neck between two powder particles by elemental diffusion and solidification, which is the reason for the increase in density. The evaporation of the elements, may be due to the fact that too high a temperature can cause overburning.As for the slight decrease in our density it could be mainly due to the error in the measurement process. For the decrease in hardness, we found that it still showed this trend after remeasurement, for which we were also puzzled and did not find the exact cause. We have therefore made some changes in the manuscript, as follows:A possible explanation for this density and hardness increase may be related to the fact that plasma was formed between the powder particles during the SPS sintering process using high-frequency discharge, which can establish a sintering neck between two powder particles via element diffusion and solidification.

Point 10: What the image at the bottom right of figue 3 shows?

Response 10: Thank you very much for your good suggestions.The image at the bottom right of Figure 3 shows the shape of the specimen before compression and after compression fracture.

Point 11: Images 4a and b are the same, correct it. And correct the text.

Response 11: Thank you very much for your good suggestions.Images 4a and b have been corrected. 

Point 12: There is no explanation of image 5c, add it.

Response 12: Thank you very much for your good suggestions.We have revised it.

Point 13: The Zr spectrum in image 6 is not understandable, correct it.

Response 13: Thank you very much for your good suggestions. According to your suggestion, we revised XRD patterns.

Point 14: Include a graph with the evolution of grain size according to temperature.

Response 14: Thank you very much for your good suggestions. Sorry, because we didn't consider this at that time.,we didn't have graph of grain size.

Point 15: Include a graph with the sintering process, where the heating process, force application and piston displacement depending on temperature are shown.

Response 15: Thank you very much for your good suggestions.Sorry, we didn't save the sintering curve data, Only part of the screenshots are saved, so they are not placed in the manuscript.

Point 16: The English must be corrected.

Response 16: Thank you very much for your suggestion .According to your suggestion, we revised the manuscript carefully and tried to minimize typographical and grammatical error.

Point 17: References 12 and 24 are the same.

Response 17: Thank you very much for your suggestion. We have carefully checked and revised it.

From your comments, it’s obvious that you are an expert. Moreover, we revised some sentences in order to make the manuscript more concise. We hope that the language is now acceptable for publication. Thank you again for your time and consideration.

We appreciate for Editors/Reviewers’ comments and suggestions earnestly. we corrected some expression errors, and hope that the corrections will meet with your approval.

Thank you again for your time and consideration.

    We look forward to your reply to our revised manuscript.

Yours sincerely,

FENGXIAN LI

Article File with Track CHanges,Please see the attachment.

Round 2

Reviewer 1 Report

The revised version answer the main concerns that I previously comment.

However, there is no answer on how the elastic modulus was measured. I understand that maybe the authors are not specialists in mechanics but they can ask for advices to colleagues. Thus, the measure of the Elastic modulus in compression tests is tricky and almost impossible without special equipment. The samples are relatively short, the authors probably did not used extensometers to measure the strain, thus the strain was probably deduced from the basic testing machine transverse beam displacement and divided by the initial length. Is that the case ? If so, it probably means that the elasticity of the testing machine itself is measured along with the elasticity of the sample. The intrinsic elasticity of the machine can be determined by proceeding to compression tests on known materials. 

So, the authors should precise how the strain of the sample was measures, if it was measured directly on the sample with an extensometer. If this is not the case the authors should eliminate the Young modulus measurements from your article, since without supplementary tests to qualify the intrinsic testing machine stiffness highly influences the deduced Young modulus values. But maybe the authors can describe how they properly taken into account the intrinsic stiffness of the testing machine, which needs a section in the article to specify how they proceed. Please ask to a mechanics specialist for mode details.

Thus I recommend minor revision to this article. 

Author Response

Point 1:The revised version answer the main concerns that I previously comment.

However, there is no answer on how the elastic modulus was measured. I understand that maybe the authors are not specialists in mechanics but they can ask for advices to colleagues. Thus, the measure of the Elastic modulus in compression tests is tricky and almost impossible without special equipment. The samples are relatively short, the authors probably did not used extensometers to measure the strain, thus the strain was probably deduced from the basic testing machine transverse beam displacement and divided by the initial length. Is that the case ? If so, it probably means that the elasticity of the testing machine itself is measured along with the elasticity of the sample. The intrinsic elasticity of the machine can be determined by proceeding to compression tests on known materials.

So, the authors should precise how the strain of the sample was measures, if it was measured directly on the sample with an extensometer. If this is not the case the authors should eliminate the Young modulus measurements from your article, since without supplementary tests to qualify the intrinsic testing machine stiffness highly influences the deduced Young modulus values. But maybe the authors can describe how they properly taken into account the intrinsic stiffness of the testing machine, which needs a section in the article to specify how they proceed. Please ask to a mechanics specialist for mode details.

Thus I recommend minor revision to this article.

Response 1: We apologize for taking so long to answer your comments. Thank you very much for your patience in suggesting the shortcomings of our article. We consulted an expert in this field at your suggestion, and the Young's modulus we obtained using compression is indeed not accurate. Therefore, we decided to remove the Young's modulus from the article. Thank you again for your suggestion, it was very useful to us and made us more rigorous in our next study.

In addition to this, we also revised the article by adding a sintering curve,as shown in Figure 1.

Article File with Track CHanges.1,Please see the attachment.

From your comments, it’s obvious that you are an expert. Moreover, we revised some sentences in order to make the manuscript more concise. We hope that the language is now acceptable for publication. Thank you again for your time and consideration.

We appreciate for Editors/Reviewers’ comments and suggestions earnestly. we corrected some expression errors, and hope that the corrections will meet with your approval.

Thank you again for your time and consideration.

    We look forward to your reply to our revised manuscript.

Yours sincerely,

FENGXIAN LI

Reviewer 2 Report

Reviewing the responses to my earlier comments, I hoped to find an improved manuscript, which would be nice and easy to read. Unfortunately, I found a material with many flaws and also, the lack of interest of the authors to correct my comments is appreciated.

Even so, I took the trouble to review the manuscript again and write my comments again.

I consider that if these comments are not corrected, this manuscript should be rewritten and re-uploaded for its publication.

- According to my earlier comment number 1, the authors responded “…we revised the manuscript carefully and tried to minimize typographical and spelling error.” Even so, in the manuscript there are errors that cannot be allowed. Review the entire manuscript and correct any errors.

- To my comment number 2 “I consider that the text of section 2.1 should go in the introduction” the authors replied “Thank you very much for your good suggestions. We think your suggestion is very good, but after we have thought about it, we feel that it is necessary to highlight our production process in this section, so it is better to write it as a separate section”. The authors indicate that it is necessary to emphasize the sintering method, but when describing it they emit very important information, such as the description of the pulses used (ton, toff parameters), the pulse frequency, the sintering graph and etc. Without this information, it would be impossible to repeat the sintering process of this material in other laboratories. For this reason, I considered that the authors should add detailed information on the sintering process (pulse current parameters, sintering curve and etc.), so that this manuscript can be published.

- Enter the metal jar measurements in the text.

- The answer to my comment number 6 is incorrect. It is necessary to indicate the heating rate as 100K/min, and not in the way proposed by the authors, which is incorrect.

- The authors have not given a clear answer to my comment number 7. Please describe the method used for the determination of the Fracture stress. The size of the samples is not indicated.

- The text “which can establish a sintering neck between two powder particles” must be demonstrated with SEM-images of the material structure.

- Please, indicate the sizes of the samples used for the compression fracture determination.

- From the text “Obviously, black clusters in the structure can be observed when the sintering temperatures are 1223 K and 1323 K, as shown in Figure. 5(a), (b) and (c).” It is not understood at what temperatures the samples in the Figure images were sintered. 5(a), (b) and (c).

- My earlier comment number 13 refers to the spectrum of figure 6 and not of figure 7.

- The lack of the sintering curve data is a big miss for this manuscript.

The authors must include this sintering curve, so that the article can be accepted. If this is not possible, the manuscript should be declined, the missing information added, and resubmitted for correction.

Author Response

Response to Reviewer2 Comments

Point 1:According to my earlier comment number 1, the authors responded “…we revised the manuscript carefully and tried to minimize typographical and spelling error.” Even so, in the manuscript there are errors that cannot be allowed. Review the entire manuscript and correct any errors.

Response 1: Thank you for your kind reminder that we have rechecked the manuscript and done our best not to make mistakes.

Point 2:To my comment number 2 “I consider that the text of section 2.1 should go in the introduction” the authors replied “Thank you very much for your good suggestions. We think your suggestion is very good, but after we have thought about it, we feel that it is necessary to highlight our production process in this section, so it is better to write it as a separate section”. The authors indicate that it is necessary to emphasize the sintering method, but when describing it they emit very important information, such as the description of the pulses used (ton, toff parameters), the pulse frequency, the sintering graph and etc. Without this information, it would be impossible to repeat the sintering process of this material in other laboratories. For this reason, I considered that the authors should add detailed information on the sintering process (pulse current parameters, sintering curve and etc.), so that this manuscript can be published.

Response 2: Thank you for your patience. We have adjusted the structure of the article based on your previous comments and have added the missing sintering curves to the manuscript.

Point 3:Enter the metal jar measurements in the text.

Response 3: Thank you for the reminder that we have added the dimensions of the metal jar to the manuscript.

Point 4:The answer to my comment number 6 is incorrect. It is necessary to indicate the heating rate as 100K/min, and not in the way proposed by the authors, which is incorrect.

Response 4: Thank you for your patience and guidance! We have revised it.

Point 5:The authors have not given a clear answer to my comment number 7. Please describe the method used for the determination of the Fracture stress. The size of the samples is not indicated.

Response 5: According to the national standard GB/T7314-2005, the compression specimen is cut into a cylindrical specimen of size Ø 2.5 mm × 5 mm to ensure uniform axial force on the specimen during compression, the compression rate is 0.1 mm/min, and the stress generated when the specimen is compressed to fracture is the fracture stress. The dimensions of the samples have been mentioned in Section 2.3 and we have marked them in red.

Point 6:The text “which can establish a sintering neck between two powder particles” must be demonstrated with SEM-images of the material structure.

Response 6: Thank you very much for your advice to us. The generation of sintered necks is a fundamental principle of powder metallurgy and has been confirmed in previous studies, which were not previously marked here due to our error, as shown in Ref. 24.

Point 7:Please, indicate the sizes of the samples used for the compression fracture determination.

Response 7: The dimensions of the compressed sample we have been explained in Section 2.3, and are marked in red.

Point 8:From the text “Obviously, black clusters in the structure can be observed when the sintering temperatures are 1223 K and 1323 K, as shown in Figure. 5(a), (b) and (c).” It is not understood at what temperatures the samples in the Figure images were sintered. 5(a), (b) and (c).

Response 8: Thank you for reminding us, this is our mistake and it has been corrected.

Point 9:My earlier comment number 13 refers to the spectrum of figure 6 and not of figure 7.

Response 9: We are sorry for this and have corrected it.

Point 10:The lack of the sintering curve data is a big miss for this manuscript.

Response 10: Thank you very much for your suggestion, we have added the sintering curve to the manuscript, as shown in Figure 1.

Article File with Track CHanges.2,Please see the attachment.

Thank you again for your time and consideration.

We appreciate for Reviewers’ comments and suggestions earnestly. we corrected some expression errors, and hope that the corrections will meet with your approval.

Thank you again for your time and consideration.

    We look forward to your reply to our revised manuscript.

Yours sincerely,

FENGXIAN LI

Round 3

Reviewer 2 Report

It is necessary to choose the type of manuscript (line 1) and correct the number in figure 45 (line 204).

After these corrections the article can be published.

Author Response

Response to Reviewer2 Comments

Point 1:It is necessary to choose the type of manuscript (line 1) and correct the number in figure 45 (line 204).

After these corrections the article can be published.

Response 1: Thank you very much for your suggestion. We have revised it in the manuscript.

Article File with Track CHanges,Please see the attachment.

Thank you again for your time and consideration.

We appreciate for Reviewers’ comments and suggestions earnestly. We have corrected some mistakes,and hope that the corrections will meet with your approval.

Yours sincerely,

FENGXIAN LI